# Metric Learning for Temporal Sequence Alignment

**Damien Garreau** [*][†]
ENS
damien.garreau@ens.fr

**Rémi Lajugie** [*][†]
INRIA
remi.lajugie@inria.fr

**Sylvain Arlot** [†]
CNRS
sylvain.arlot@ens.fr

**Francis Bach** [†]
INRIA
francis.bach@inria.fr

## Abstract

In this paper, we propose to learn a Mahalanobis distance to perform alignment of multivariate time series. The learning examples for this task are time series for which the true alignment is known. We cast the alignment problem as a structured prediction task, and propose realistic losses between alignments for which the optimization is tractable. We provide experiments on real data in the audio-to-audio context, where we show that the learning of a similarity measure leads to improvements in the performance of the alignment task. We also propose to use this metric learning framework to perform feature selection and, from basic audio features, build a combination of these with better alignment performance.

## 1 Introduction

The problem of aligning temporal sequences is ubiquitous in applications ranging from bioinformatics [5, 1, 23] to audio processing [4, 6]. The goal is to align two similar time series that have the same global structure, but local temporal differences. Most alignments algorithms rely on similarity measures, and having a good metric is crucial, especially in the high-dimensional setting where some features of the signals can be irrelevant to the alignment task. The goal of this paper is to show how to learn this similarity measure from annotated examples in order to improve the relevance of the alignments.

For example, in the context of music information retrieval, alignment is used in two different cases: (1) audio-to-audio alignment and (2) audio-to-score alignment. In the first case, the goal is to match two audio interpretations of the same piece that are potentially different in rythm, whereas audio-to-score alignment focuses on matching an audio signal to a symbolic representation of the score. In the second case, there are some attempts to learn from annotated data a measure for performing the alignment. Joder et al. [12] propose to fit a generative model in that context, and Keshet et al. [13] learn this measure in a discriminative setting.

Similarly to Keshet et al. [13], we use a discriminative loss to learn the measure, but our work focuses on audio-to-audio alignment. In that context, the set of authorized alignments is much larger, and we explicitly cast the problem as a structured prediction task, that we solve using off-the-shelf stochastic optimization techniques [15] but with proper and significant adjustments, in particular in terms of losses. The ideas of alignment are also very relevant to the community of speech recognition since the pioneering work of Sakoe and Chiba [19].

---

[*]Contributed equally
[†]SIERRA project-team, Département d'Informatique de l'Ecole Normale Supérieure (CNRS, INRIA, ENS)

The need for metric learning goes far beyond unsupervised partitioning problems. Weinberger and Saul [26] proposed a large-margin framework for learning a metric in nearest-neighbour algorithms based on sets of must-link/must-not-link constraints. Lajugie et al. [16] proposed to use a large margin framework to learn a Mahalanobis metric in the context of partitioning problems. Since structured SVM have been proposed by Tsochantaridis et al. [25] and Taskar et al. [22], they have successfully been used to solve many learning problems, for instance to learn weights for graph matching [3] or a metric for ranking tasks [17]. They have also been used to learn graph structures using graph cuts [21].

We make the following five contributions:

– We cast the learning of a Mahalanobis metric in the context of alignment as a structured prediction problem.
– We show that on real musical datasets this metric improves the performance of alignment algorithms using high-level features.
– We propose to use the metric learning framework to learn combinations of basic audio features and get good alignment performances.
– We show experimentally that the standard Hamming loss, although tractable computationnally, does not permit to learn a relevant similarity measure in some real world settings.
– We propose a new loss, closer to the true evaluation loss for alignments, leading to a tractable learning task, and derive an efficient Frank-Wolfe-based algorithm to deal with this new loss. That loss solves some issues encountered with the Hamming loss.

## 2 Matricial formulation of alignment problems

### 2.1 Notations

In this paper, we consider the alignment problem between two multivariate time series sharing the same dimension $p$, but possibly of different lengths $T_A$ and $T_B$, namely $A \in \mathbb{R}^{T_A \times p}$ and $B \in \mathbb{R}^{T_B \times p}$. We refer to the rows of $A$ as $a_1, \ldots, a_{T_A} \in \mathbb{R}^p$ and those of $B$ as $b_1, \ldots, b_{T_B} \in \mathbb{R}^p$ as column vectors. From now on, we denote by $X$ the pair of signals $(A, B)$.

Let $C(X) \in \mathbb{R}^{T_A \times T_B}$ be an arbitrary pairwise *affinity matrix* associated to the pair $X$, that is, $C(X)_{i,j}$ encodes the affinity between $a_i$ and $b_j$. Note that our framework can be extended to the case where $A$ and $B$ are multivariate signals of different dimensions, as long as $C(X)$ is well-defined. The goal of the alignment task is to find two non-decreasing sequences of indices $\alpha$ and $\beta$ of same length $u \geq \max(T_A, T_B)$ and to match each time index $\alpha(i)$ in the time series $A$ to the time index $\beta(i)$ in the time series $B$, in such a way that $\sum_{i=1}^{u} C(X)_{\alpha(i),\beta(i)}$ is maximal, and that $(\alpha, \beta)$ satisfies:

$$
\begin{cases}
\alpha(1) = \beta(1) = 1 & \text{(matching beginnings)} \\
\alpha(u) = T_A, \beta(u) = T_B & \text{(matching endings)} \\
\forall i, (\alpha(i+1), \beta(i+1)) - (\alpha(i), \beta(i)) \in \{(1,0),(0,1),(1,1)\} & \text{(three type of moves)}
\end{cases}
\tag{1}
$$

For a given $(\alpha, \beta)$, we define the binary matrix $Y \in \{0,1\}^{T_A \times T_B}$ such that $Y_{\alpha(i),\beta(i)} = 1$ for every $i \in \{1, \ldots, u\}$ and 0 otherwise. We denote by $\mathcal{Y}(X)$ the set of such matrices, which is uniquely determined by $T_A$ and $T_B$. An example is given in Fig. 1. A vertical move in the $Y$ matrix means that the signal $B$ is waiting for $A$, whereas an horizontal one means that $A$ is waiting for $B$, and a diagonal move means that they move together. In this sense the time reference is "warped".

When $C(X)$ is known, the alignment task can be cast as the following linear program (LP) over the set $\mathcal{Y}(X)$:

$$
\max_{Y \in \mathcal{Y}(X)} \text{Tr}(C(X)^\top Y).
\tag{2}
$$

Our goal is to learn how to form the affinity matrix: once we have learned $C(X)$, the alignment is obtained from Eq. (2). The optimization problem in Eq. (2) will be referred to as the *decoding* of our model.

**Dynamic time warping.** Given the affinity matrix $C(X)$ associated with the pair of signals $X = (A, B)$, finding the alignment that solves the LP of Eq. (2) can be done efficiently in $O(T_A T_B)$ using

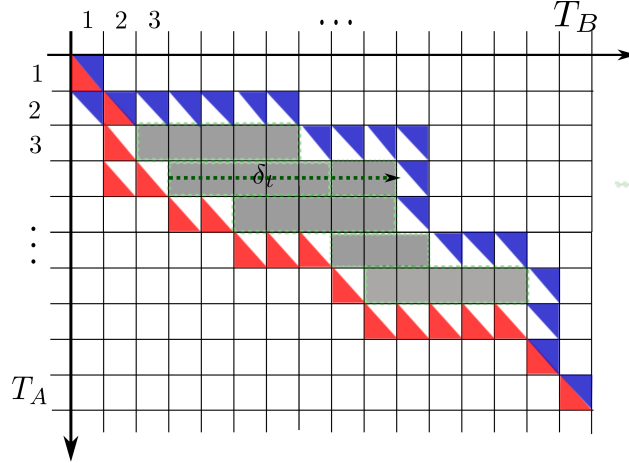

Figure 1: Example of two valid alignments encoded by matrices $Y^1$ and $Y^2$. Red upper triangles show the $(i,j)$ such that $Y^1_{i,j} = 1$, and the blue lower ones show the $(i,j)$ such that $Y^2_{i,j} = 1$. The grey zone corresponds to the area loss $\delta_{\mathrm{abs}}$ between $Y^1$ and $Y^2$.

a dynamic programming algorithm. It is often referred to as dynamic time warping [5, 18]. This algorithm is described in Alg. 1 of the supplementary material. Various additional constraints may be used in the dynamic time warping algorithm [18], which we could easily add to Alg. 1.

The cardinality of the set $\mathcal{Y}(X)$ is huge: it corresponds to the number of paths on a rectangular grid from the southwest $(1,1)$ to the northeast corner $(T_A, T_B)$ with vertical, horizontal and diagonal moves allowed. This is the definition of the Delannoy numbers [2]. As noted in [24], when $t = T_A = T_B$ goes to infinity, and one can show that $\#\mathcal{Y}_{t,s} \sim \frac{(3+2\sqrt{2})^t}{\sqrt{\pi t}\sqrt{3\sqrt{2}-4}}$.

## 2.2 The Mahalanobis metric

In many applications (see, e.g., [6]), for a pair $X = (A, B)$, the affinity matrix is computed by $C(A,B)_{i,j} = -\|a_{i,k} - b_{j,k}\|^2$. In this paper we propose to learn the metric to compare $a_i$ and $b_j$ instead of using the plain Euclidean metric. That is, $C(X)$ is parametrized by a matrix $W \in \mathcal{W} \subset \mathbb{R}^{p \times p}$, where $\mathcal{W} \subset \mathbb{R}^{p \times p}$ is the set of semi-definite positive matrices, and we use the corresponding Mahalanobis metric to compute the pairwise affinity between $a_i$ and $b_j$:

$$C(X;W)_{i,j} = -(a_i - b_j)^\top W (a_i - b_j). \tag{3}$$

Note that the decoding of Eq. (2) is the maximization of a linear function in the parameter $W$:

$$\max_{Y \in \mathcal{Y}(X)} \mathrm{Tr}(C(X;W)^\top Y) \quad \Leftrightarrow \quad \max_{Y \in \mathcal{Y}(X)} \mathrm{Tr}(W^\top \phi(X,Y)), \tag{4}$$

if we define the joint feature map

$$\phi(X,Y) = -\sum_{i=1}^{T_A} \sum_{j=1}^{T_B} Y_{i,j}(a_i - b_j)(a_i - b_j)^\top \in \mathbb{R}^{p \times p}. \tag{5}$$

## 3 Learning the metric

From now on, we assume that we are given $n$ pairs of training instances[1] $(X^i, Y^i) = ((A^i, B^i), Y^i) \in \mathbb{R}^{T_A^i \times p} \times \mathbb{R}^{T_B^i \times p} \times \{0,1\}^{T_A^i \times T_B^i}$, $i = 1, \ldots, n$. Our goal is to find a matrix $W$ such that the predicted alignments are close to the groundtruth on these examples, as well as on unseen examples. We first define a *loss* between alignments, in order to quantify the proximity between alignments.

### 3.1 Losses between alignments

In our framework, the alignments are encoded by matrices in $\mathcal{Y}(X)$, thus we are interested in functions $\ell : \mathcal{Y}(X) \times \mathcal{Y}(X) \to \mathbb{R}_+$. The Frobenius norm is defined by $\|M\|_F^2 = \sum_{i,j} M_{i,j}^2$.

**Hamming loss.** A simple loss between matrices is the Frobenius norm of their difference, which turns out to be the unnormalized Hamming loss [9] for 0/1-valued matrices. For two matrices $Y_1, Y_2 \in \mathcal{Y}(X)$, it is defined as:

$$\ell_H(Y_1, Y_2) = \|Y_1 - Y_2\|_F^2 = \mathrm{Tr}(Y_1^\top Y_1) + \mathrm{Tr}(Y_2^\top Y_2) - 2\,\mathrm{Tr}(Y_1^\top Y_2)$$
$$= \mathrm{Tr}(Y_1 \mathbf{1}_{T_B} \mathbf{1}_{T_A}^\top) + \mathrm{Tr}(Y_2 \mathbf{1}_{T_B} \mathbf{1}_{T_A}^\top) - 2\,\mathrm{Tr}(Y_1^\top Y_2), \tag{6}$$

where $\mathbf{1}_T$ is the vector of $\mathbb{R}^T$ with all coordinates equal to 1. The last line of Eq. (6) comes from the fact that the $Y_i$ have 0/1-values; that makes the Hamming loss affine in $Y_1$ and $Y_2$. This loss is often used in other structured prediction tasks [15]; in the audio-to-score setting, Keshet et al. [13] use a modified version of this loss, which is the average number of times the difference between the two alignments is greater than a fixed threshold.

This loss is easy to optimize since, it is linear in our parametrization of the alignement problem, but not optimal for audio-to-audio alignment. Indeed, a major drawback of the Hamming loss is that, for alignments of fixed length, it depends only on the number of "crossings" between alignment paths: one can easily find $Y_1, Y_2, Y_3$ such that $\ell_H(Y_2, Y_1) = \ell_H(Y_3, Y_1)$ but $Y_2$ is much closer to $Y_1$ than $Y_3$ (see Fig. 2). It is important to notice this is often the case when the length of the signals grows.

**Area loss.** A more natural loss can be computed as the mean distance beween the paths depicted by two matrices $Y^1, Y^2 \in \mathcal{Y}(X)$. This loss corresponds to the area between the paths of two matrices $Y$, as represented by the grey zone on Fig. 1.

Formally, as in Fig. 1, for each $t \in \{1, \ldots, T_B\}$ we put $\delta_t = |\min\{k, Y_{t,k}^1 = 1\} - \min\{k, Y_{t,k}^2 = 1\}|$. Then the area loss is the mean of the $\delta_t$. In the audio literature [14], this loss is sometimes called the "mean absolute deviation" loss and is noted $\delta_{\mathrm{abs}}(Y^1, Y^2)$.

Unfortunately, for the general alignment problem, $\delta_{\mathrm{abs}}$ is not linear in the matrices $Y$. But in the context of alignment of sequences of two different natures, one of the signal is a reference and thus the index sequence $\alpha$ defined in Eq. (1) is increasing, e.g., for the audio-to-partition alignment problem [12]. This loss is then linear in each of its arguments. More precisely, if we introduce the matrix $L_{T_A} \in \mathbb{R}^{T_A \times T_A}$ which is lower triangular with ones (including on the diagonal), we can write the loss as

$$\ell_O = \|L_{T_A}(Y_1 - Y_2)\|_F^2 \tag{7}$$
$$= \mathrm{Tr}(L_{T_A} Y_1 \mathbf{1}_{T_B} \mathbf{1}_{T_A}^\top) + \mathrm{Tr}(L_{T_A} Y_2 \mathbf{1}_{T_B} \mathbf{1}_{T_A}^\top) - 2\,\mathrm{Tr}(L_{T_A} Y_1 Y_2^\top L_{T_A}^\top).$$

We now prove that this loss corresponds to the area loss in this special case. Let $Y$ be an alignment, then it is easy see that $(L_{T_A} Y)_{i,j} = \sum_k (L_{T_A})_{i,k} Y_{k,j} = \sum_{k=1}^i Y_{k,j}$. If $Y$ does not have vertical moves, i.e., for each $j$ there is an unique $k_j$ such that $Y_{k_j,j} = 1$, we have that $(L_{T_A} Y)_{i,j} = 1$ if and only if $i \geq k_j$. So $\sum_{i,j} (L_{T_A} Y)_{i,j} = \#\{(i,j), i \geq k_j\}$, which is exactly the area under the curve determined by the path of $Y$. In all our experiments, we use $\delta_{\mathrm{abs}}$ for evaluation but not for training.

**Approximation of the area loss: the symmetrized area loss.** In many real world applications [14], a meaningful loss to assess the quality of an alignment is the area loss. As shown by our experiments, if the Hamming loss is sufficient in some simple situations and allows to learn a metric that leads to good alignment performance in terms of area loss, on more challenging datasets it does not work at all (see Sec. 5). This is due to the fact that two alignments that are very close in terms of area loss can suffer a big Hamming loss (cf. Fig. 2). Thus it is natural to extend the formulation of Eq. (7) to matrices in $\mathcal{Y}(X)$. We start by symmetrizing the formulation of Eq. (7) to overcome problems of overpenalization of vertical vs. horizontal moves. We define, for any couple of binary matrices $(Y^1, Y^2)$,

$$\ell_S(Y_1, Y_2) = \frac{1}{2}\big(\|L_{T_A}(Y_1 - Y_2)\|_F^2 + \|(Y_1 - Y_2)L_{T_B}\|_F^2\big) \tag{8}$$
$$= \frac{1}{2}\Big[\mathrm{Tr}(Y_1^\top L_{T_A}^\top L_{T_A} Y_1) + \mathrm{Tr}(L_{T_A} Y_2 \mathbf{1}_{T_B} \mathbf{1}_{T_A}^\top) - 2\,\mathrm{Tr}(Y_2^\top L_{T_A}^\top L_{T_A} Y_1)$$
$$+ \mathrm{Tr}(Y_1 L_{T_B} L_{T_B}^\top Y) + \mathrm{Tr}(Y_2^\top \mathbf{1}_{T_A} \mathbf{1}_{T_B}^\top L_{T_B} L_{T_B}^\top Y_2) - 2\,\mathrm{Tr}(Y_2 L_{T_B} L_{T_B}^\top Y_1^\top)\Big].$$

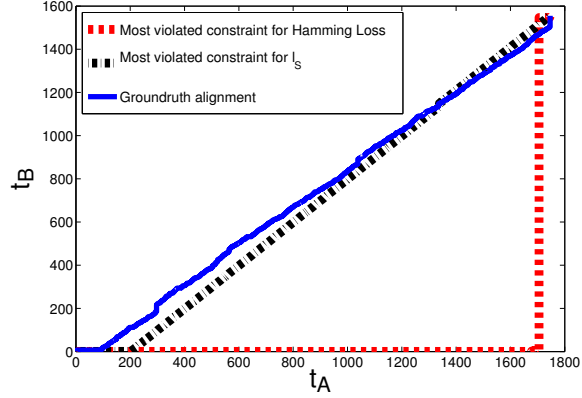

Figure 2: On the real world Bach chorales dataset, we have represented a groundtruth alignment together with two others. In term of Hamming loss, both alignments are as far from the groundtruth whereas for the area loss, they are not. In the structured prediction setting described in Sec. 4, the depicted alignment are the so-called "most violated constraint", namely the output of the loss augmented decoding step (see Sec. 4).

We propose now to make this loss *concave* over the convex hull of $\mathcal{Y}(X)$ that we denote from now on $\overline{\mathcal{Y}}(X)$. Let us introduce $D_T = \lambda_{\max}(L_T^\top L_T)I_{T \times T}$ with $\lambda_{\max}(U)$ the largest eigenvalue of $U$. [2] For any binary matrices $Y_1$, $Y_2$, we have

$$
\begin{aligned}
\ell_S(Y_1, Y_2) = \frac{1}{2} \Big[ &\operatorname{Tr}(Y_1^\top (L_{T_A}^\top L_{T_A} - D_{T_A})Y_1) + \operatorname{Tr}(D_{T_A}Y_1\mathbf{1}_{T_B}\mathbf{1}_{T_A}^\top) + \operatorname{Tr}(L_{T_A}Y_2\mathbf{1}_{T_B}\mathbf{1}_{T_A}^\top) \\
&- 2\operatorname{Tr}(Y_2^\top (L_{T_A}^\top L - D_{T_A})Y_1) + \operatorname{Tr}(Y_1(L_{T_B}L_{T_B}^\top - D_{T_B})Y) \\
&+ \operatorname{Tr}(Y_1 D_{T_B}\mathbf{1}_{T_B}\mathbf{1}_{T_A}^\top) + \operatorname{Tr}(Y_2^\top L_{T_B}L_{T_B}^\top Y_2) - 2\operatorname{Tr}(Y_2 L_{T_B}L_{T_B}^\top Y_1^\top) \Big],
\end{aligned}
$$

and we get a *concave* function over $\overline{\mathcal{Y}}(X)$ that coincides with $\ell_S$ on $\mathcal{Y}(X)$.

### 3.2 Empirical loss minimization

Recall that we are given $n$ alignment examples $(X^i, Y^i)_{1 \leq i \leq n}$. For a fixed loss $\ell$, our goal is now to solve the following minimization problem in $W$:

$$
\min_{W \in \mathcal{W}} \left\{ \frac{1}{n} \sum_{i=1}^n \ell\Big(Y^i, \operatorname*{argmax}_{Y \in \mathcal{Y}_{T_A^i, T_B^i}} \operatorname{Tr}(C(X^i; W)^\top Y)\Big) + \lambda \Omega(W) \right\}, \tag{9}
$$

where $\Omega = \frac{\lambda}{2}\|W\|_F^2$ is a convex regularizer preventing from overfitting, with $\lambda \geq 0$.

## 4 Large margin approach

In this section we describe a large margin approach to solve a surrogate to the problem in Eq. (9), which is untractable. As shown in Eq. (4), the decoding task is the maximum of a linear function in the parameter $W$ and aims at predicting an output over a large and discrete space (the space of potential alignments with respect to the constraints in Eq. (1)). Learning $W$ thus falls into the structured prediction framework [25, 22]. We define the hinge loss, a convex surrogate, by

$$
L(X, Y; W) = \max_{Y' \in \mathcal{Y}(X)} \left\{ \ell(Y, Y') - \operatorname{Tr}(W^\top [\phi(X, Y) - \phi(X, Y')]) \right\}. \tag{10}
$$

The evaluation of $L$ is usually referred to as "loss-augmented decoding", see [25]. If we define $\widehat{Y}^i$ as the argmax in Eq. (10) when $(X, Y) = (X^i, Y^i)$, then elementary computations show that

$$\widehat{Y}^i = \operatorname*{argmin}_{Y \in \mathcal{Y}(X)} \operatorname{Tr}((U^\top - 2Y^{i\top} - C(X^i; W)^\top)Y),$$

where $U = \mathbf{1}_{T_B} \mathbf{1}_{T_B}^\top \in \mathbb{R}^{T_A \times T_B}$.

We now aim at solving the following problem, sometimes called the *margin-rescaled problem*:

$$\min_{W \in \mathcal{W}} \frac{\lambda}{2} \|W\|_F^2 + \frac{1}{n} \sum_{i=1}^n \max_{Y \in \mathcal{Y}(X)} \left\{ \ell(Y, Y^i) - \operatorname{Tr}(W^\top \left[ \phi(X^i, Y^i) - \phi(X^i, Y) \right]) \right\}. \qquad (11)$$

**Hamming loss case.** From Eq. (4), one can notice that our joint feature map is linear in $Y$. Thus, if we take a loss that is linear in the first argument of $\ell$, for instance the Hamming loss, the loss-augmented decoding is the maximization of a linear function over the spaces $\mathcal{Y}(X)$ that we can solve efficiently using dynamic programming algorithms (see Sec. 2.1 and supplementary material).

That way, plugging the Hamming loss (Eq. (6)) in Eq. (11) leads to a convex structured prediction problem. This problem can be solved using standard techniques such as cutting plane methods [11], stochastic gradient descent [20], or block-coordinate Frank-Wolfe in the dual [15]. Note that we adapted the standard unconstrained optimization methods to our setting, where $W \succeq 0$.

**Optimization using the symmetrized area loss.** The symmetrized area loss is concave in its first argument, thus the problem of Eq. (11) is in a min/max form and deriving a dual is straightforward. Details can be found in the supplementary material. If we plug the symmetrized area loss $\ell_S$ (SAL) defined in Eq. (8) into our problem (11), we can show that the dual of (11) has the following form:

$$\min_{(Z^1, \ldots, Z^n) \in \overline{\mathcal{Y}}} \frac{1}{2\lambda n^2} \| \sum_{i=1}^n - \sum_{j,k} (Y_i - Z^i)_{j,k}(a_j - b_k)(a_j - b_k)^T \|_F^2 - \frac{1}{n} \sum_{i=1}^n \ell_S(Z, Z^i), \quad (12)$$

if we denote by $\overline{\mathcal{Y}}(X^i)$ the convex hull of the sets $\mathcal{Y}(X^i)$, and by $\overline{\mathcal{Y}}$ the cartesian product over all the training examples $i$ of such sets. Note that we recover a similar result as [15]. Since the SAL loss is concave, the aforementioned problem is convex.

The problem (12) is a quadratic program over the compact set $\overline{\mathcal{Y}}$. Thus we can use a Frank-Wolfe [7] algorithm. Note that it is similar to the one proposed by Lacoste-Julien et al. [15] but with an additional term due to the concavity of the loss.

## 5   Experiments

We applied our method to the task of learning a good similarity measure for aligning audio signals. In this field researchers have spent a lot of efforts in designing well-suited and meaningful features [12, 4]. But the problem of combining these features for aligning temporal sequences is still challenging. For simplicity, we took $W$ diagonal for our experiments.

### 5.1   Dataset of Kirchhoff and Lerch [14]

**Dataset description.** First, we applied our method on the dataset of Kirchhoff and Lerch [14]. In this dataset, pairs of aligned examples $(A^i, B^i)$ are artificially created by stretching an original audio signal. That way, the groundtruth alignment $Y^i$ is known and thus the data falls into our setting A more precise description of the dataset can be found in [14].

The $N = 60$ pairs are stretched along two different tempo curves. Each signal is made of 30s of music divided in frames of 46ms with a hopsize of 23ms, thus leading to a typical length of the signals of $T \approx 1300$ in our setting. We keep $p = 11$ features that are simple to implement and known to perform well for alignment tasks [14]. Those were: five MFCC [8] (labeled $M1, \ldots, M5$ in Fig. 3), the spectral flatness (SF), the spectral centroid (SC), the spectral spread (SS), the maximum of the envelope (Max), and the power level of each frame (Pow), see [14] for more details on the computation of the features. We normalize each feature by subtracting the median value and dividing by the standard deviation to the median, as audio data are subject to outliers.

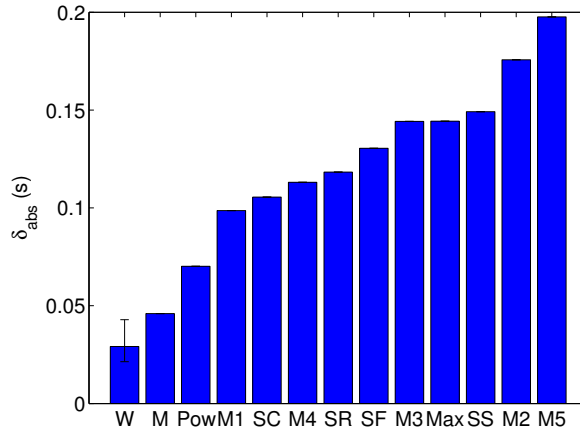

Figure 3: Comparison of performance between individual features and the learned metric. Error bars for the performance of the learned metric were determined with the best and the worst performance on 5 different experiments. $W$ denotes the learned combination using our method, and $M$ the best MFCC combination.

**Experiments.** We conducted the following experiment: for each individual feature, we perform alignment using dynamic time warping algorithm and evaluate the performance of this single feature in terms of losses typically used to asses performance in this setting [14]. In Fig. 3, we report the results of these experiments.

Then, we plug these data into our method, using the Hamming loss to learn a linear positive combination of these features. The result is reported in Fig. 3. Thus, combining these features on this dataset yields to better performances than only considering a single feature.

For completeness, we also conducted the experiments using the standard 13 first MFCCs coefficients and their first and second order derivatives as features. These results competed with the best learned combination of the handcrafted features. Namely, in terms of the $\delta_{\mathrm{abs}}$ loss, they perform at $0.046$ seconds. Note that these results are slightly worse than the best single handcrafted feature, but better than the best MFCC coefficient used as a feature.

As a baseline, we also compared ourselves against the uniform combination of handcrafted features (the metric being the identity matrix). The results are off the charts on Fig. 3 with $\delta_{\mathrm{abs}}$ at $4.1$ seconds (individual values ranging from $1.4$ seconds to $7.4$ seconds).

## 5.2 Chorales dataset

**Dataset.** The Bach 10 dataset[3] consists in ten J. S. Bach's Chorales (small quadriphonic pieces). For each Chorale, a MIDI reference file corresponding to the "score", or basically a representation of the partition. The alignments between the MIDI files and the audio file are given, thus we have converted these MIDI files into audio following what is classically done for alignment (see e.g, [10]). That way we fall into the audio-to-audio framework in which our technique apply. Each piece of music is approximately 25s long, leading to similar signal length ($T \approx 1300$).

**Experiments.** We use the same features as in Sec. 5.1. As depicted in Fig. 4, the optimization with Hamming loss performs poorly on this dataset. In fact, the best individual feature performance is far better than the performance of the learned $W$. Thus metric learning with the "practical" Hamming loss performs much worse than the best single feature.

Then, we conducted the same learning experiment with the symetrized area loss $\ell_S$. The resulting learned parameter is far better than the one learned using the Hamming loss. We get a performance that is similar to the one of the best feature. Note that these features were handcrafted and reaching their performance on this hard task with only a few training instances is already challenging.

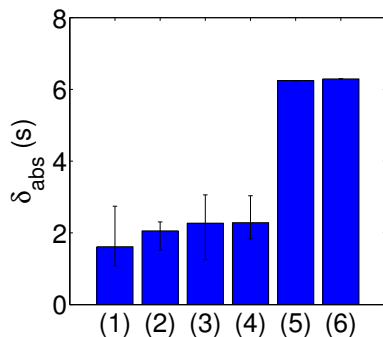

Figure 4: Performance of our algorithms on the Chorales dataset. From left to right: (1) Best single feature, (2) Best learned combination of features using the symmetrized area loss $\ell_S$, (3) Best combination of MFCC using SAL and $D_T$ obtained via SDP (see footnote in section 3) (4) Best combination of MFCC and derivatives learned with $\ell_S$, (5) Best combination of MFCCs and derivatives learned with Hamming loss, (6) Best combination of features of [14] using Hamming loss.

In Fig. 2, we have depicted the result, for a learned parameter $W$, of the loss augmented decoding performed either using the area. As it is known for structured SVM, this represents the most violated constraint [25]. We can see that the most violated constraint for the Hamming loss leads to an alignment which is totally unrelated to the groundtruth alignment whereas the one for the symmetrized area loss is far closer and much more discriminative.

## 5.3 Feature selection

Last, we conducted feature selection experiments over the same datasets. Starting from low level features, namely the 13 leading MFCCs coefficients and their first two derivatives, we learn a linear combination of these that achieves good alignment performance in terms of the area loss. Note that very little musical prior knowledge is put into these. Moreover we either improve on the best handcrafted feature on the dataset of [14] or perform similarly. On both datasets, the performance of learned combination of handcrafted features performed similarly to the combination of these 39 MFCCs coefficients.

## 6 Conclusion

In this paper, we have presented a structured prediction framework for learning the metric for temporal alignment problems. We are able to combine hand-crafted features, as well as building automatically new state-of-the-art features from basic low-level information with little expert knowledge.

Technically, this is made possible by considering a loss beyond the usual Hamming loss which is typically used because it is "practical" within a structured prediction framework (linear in the output representation).

The present work may be extended in several ways, the main one being to consider cases where only partial information about the alignments is available. This is often the case in music [4] or bioinformatics applications. Note that, similarly to Lajugie et al. [16] a simple alternating optimization between metric learning and constrained alignment provide a simple first solution, which could probably be improved upon.

**Acknowledgements.** The authors acknowledge the support of the European Research Council (SIERRA project 239993), the GARGANTUA project funded by the Mastodons program of CNRS and the Airbus foundation through a PhD fellowship. Thanks to Piotr Bojanowski, for helpful discussions. Warm thanks go to Arshia Cont and Philippe Cuvillier for sharing their knowledge about audio processing, and to Holger Kirchhoff and Alexander Lerch for their dataset.

## Footnotes

[1]We will see that it is necessary to have fully labelled instances, which means that for each pair $X^i$ we need an *exact* alignment $Y^i$ between $A^i$ and $B^i$. Partial alignment might be dealt with by alternating between metric learning and constrained alignment.

[2]For completeness, in our experiments, we also try to set the matrices $D_T$ with minimal trace that dominate $L_T^\top L_T$ by solving a semidefinite program (SDP). We report the associated result in Fig 4. Note also that other matrices could have been chosen. In particular, since our matrices $L_T$ are pointwise positive, the matrix $\operatorname{Diag}(L_T^\top L_T) - L_T^\top L_T$ is such that the loss is concave.

[3]`http://music.cs.northwestern.edu/data/Bach10.html`.

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
