[Supplementary Material]

# 1 Derivation of the BCFW-like algorithm for the quadratic loss.

## 1.1 Relaxing of the discrete optimization set for loss augmented inference.

Let us start from the global structured objective equation of the paper. Recall that we are given training examples $((X^1, Y^1), \ldots, (X^n, Y^n))$. In order to make the derivation easier, and following Lacoste-Julien et al. [3], we denote the difference between the feature map associated to any $Y \in \mathcal{Y}(X^i)$ and the one associated to the true training example label $Y_i$ by:

$$\text{Tr}(W\phi(X^i, Y^i)) = \text{Tr}(W \sum_{j,k} (Y^i_{j,k} - Y_{j,k})(a^i_j - b^i_k)(a^i_j - b^i_k)^\top) = \langle W, \psi^i(Y) \rangle, \qquad (1)$$

where $\psi(Y) = \sum_{j,k} (Y^i_{j,k} - Y_{j,k})(a^i_j - b^i_k)(a^i_j - b^i_k)^\top$. The objective of structured prediction is thus:

$$\min_{W \in \mathcal{W}} \frac{\lambda}{2} \|W\|_2^2 + \frac{1}{n} \sum_{i=1}^{n} \max_{Y \in \mathcal{Y}(X^i)} \left\{ \ell_i(Y, Y^i) - \langle W, \psi^i(Y) \rangle \right\}. \qquad (2)$$

The term $\max_{Y \in \mathcal{Y}(X^i)} \left\{ \ell_i(Y, Y^i) - \langle W, \psi^i(Y) \rangle \right\}$ corresponds to the structural hinge loss for our problem. Let us introduce $\overline{\mathcal{Y}}(X^i)$ the convex hulls of the sets $\mathcal{Y}(X^i)$. We will also make use of $\overline{\mathcal{Y}} = \overline{\mathcal{Y}}(X^1) \times \ldots \times \overline{\mathcal{Y}}(X^n)$. From now on, we perform the loss augmented decoding on this relaxed set. This problem has potentially *non integral* solutions. We call the maximization of the hinge loss over $\overline{\mathcal{Y}}$ the *loss augmented inference*, in opposition to *loss-augmented decoding* which . Now we can write a new optimization objective:

$$\min_{W \in \mathcal{W}} \frac{\lambda}{2} \|W\|_2^2 + \max_{(Z_1, \ldots, Z_n) \in \overline{\mathcal{Y}}} \left\{ \frac{1}{n} \sum_{i=1}^{n} \left[ \ell_i(Z_i, Y^i) - \langle W, \psi^i(Z_i) \rangle \right] \right\}. \qquad (3)$$

Note that since our joint feature map $\phi(X^i, Y)$ is linear in $Y$, if $\ell$ is linear as well (for instance if $\ell$ is the Hamming loss), this problem is strictly equivalent to (2) since in that case, the loss-augmented inference is a LP over $\overline{\mathcal{Y}}(X^i)$, which has necessary a solution in $\mathcal{Y}(X^i)$ (see, e.g, [Prop. B.21] of [1]).

In general, in order to be convex and thus tractable, the aforementioned problem of Eq.(3) requires a loss which is concave in the variable $Z_i$ which belong to the convex sets $\overline{\mathcal{Y}}(X^i)$.

## 1.2 Dual of the structured SVM

Since Prob. (2) is a convex optimization problem in saddle point form, we get the dual by switching the max and the min:

$$\max_{(Z_1, \ldots, Z_n) \in \overline{\mathcal{Y}}} \min_{W \in \mathcal{W}} \frac{\lambda}{2} \|W\|_2^2 + \left\{ \frac{1}{n} \sum_{i=1}^{n} \left[ \ell_i(Y, Y^i) - \langle W, \psi^i(Z_i) \rangle \right] \right\}. \qquad (4)$$

From the above equation, we deduce the following general relation linking primal variable $W$ and dual variables $(Z_1^*, \ldots, Z_n^*) \in \overline{\mathcal{Y}}(X^1) \times \ldots \times \overline{\mathcal{Y}}(X^n)$:

$$W^* \in \operatorname*{argmin}_{W \in \mathcal{W}} \frac{\lambda}{2} \|W\|_2^2 + \left\{ \frac{1}{n} \sum_{i=1}^{n} \left[ \ell_i(Y, Y^i) - \langle W, \psi^i(Z_i^*) \rangle \right] \right\}. \qquad (5)$$

Due to the isotropic form of the function in Eq.(5), this simply reduces to the Euclidean projection of the unconstrained minimum of $\frac{\lambda}{2} \|W\|_2^2 + \left\{ \frac{1}{n} \sum_{i=1}^{n} \left[ \ell_i(Y, Y^i) - \langle W, \psi^i(Z_i^*) \rangle \right] \right\}$ on $\mathcal{W}$.
In the specific case when $\mathcal{W}$ is unconstrained and simply equals to $\mathbb{R}^{p \times p}$, this reduces to:

$$W = \frac{1}{\lambda} \sum_{i=1}^{n} \psi_i(Z_i^*). \qquad (6)$$

If $\mathcal{W}$ is the set of symmetric semidefinite positive matrices we get:

$$W = \frac{1}{\lambda} \sum_{i=1}^{n} (\psi_i(Z_i^*))_+, \qquad (7)$$

with $(\psi_i(Z_i))_+$ the projection of $(\psi_i(Z_i))$ over $\mathcal{W}$.

Eventually, if we consider the set of diagonal matrices $\mathcal{D}$, and denote by $\mathrm{Diag}$ the operator associating to a matrix the matrix composed of its diagonal:

$$W = \frac{1}{\lambda} \sum_{i=1}^{n} \mathrm{Diag}(\psi_i(Z_i^*)). \tag{8}$$

These relations are also known as the "representer theorems".

For what follows we consider the case of $\mathcal{W} = \mathbb{R}^{p \times p}$ but dealing with the other cases is similar.

In that case the dual can be written simply as:

$$\max_{(Z_1,\dots,Z_n) \in \overline{\mathcal{Y}}(X^1) \times \dots \times \overline{\mathcal{Y}}(X^n)} -\frac{1}{2\lambda n^2} \| \sum_{i=1}^{n} \psi_i(Z_i) \|_F^2 + \frac{1}{n} \sum_{i=1}^{n} \ell(Y_i, Z_i). \tag{9}$$

We recover a result similar to the ones of Lacoste-Julien et al. [3].

### 1.3 A Frank-Wolfe algorithm for solving Prob. (9)

Now, we can derive a Frank-Wolfe algorithm for solving the dual problem of 9. As noted in the paper, we are able to maximize or minimize any linear form over the sets $\mathcal{Y}(X^i)$, thus we are able to solve LPs over the convex hulls $\overline{\mathcal{Y}}(X^i)$ of such sets.

Plugging back the specific form of our joint feature map directly into Eq. (9) we get that

$$\psi_i(Z^i) = -\sum_{j,k} (Y_i - Z^i)_{j,k} (a_j - b_k)(a_j - b_k)^T, \tag{10}$$

and thus we can write the dual problem as:

$$\min_{\substack{(Z^1,\dots,Z^n) \in \\ \overline{\mathcal{Y}}(X^1) \times \dots \times \overline{\mathcal{Y}}(X^n)}} \frac{1}{2\lambda n^2} \| \sum_{i=1}^{n} -\sum_{j,k} (Y_i - Z^i)_{j,k}(a_j - b_k)(a_j - b_k)^T \|_F^2 - \frac{1}{n} \sum_{i=1}^{n} \ell(Y^i, Z^i) \tag{11}$$

Now, as in the paper, let us introduce $L_{T_A} \in \mathbb{R}^{T_A \times T_A}$ and $L_{T_B} \in \mathbb{R}^{T_B \times T_B}$ (note that we omit the dependence in $i$ of $T_A$ and $T_B$). If $U_i$ is the matrix of ones of the same size as $Z^i$, we consider the following loss:

$$
\begin{aligned}
\ell(Y^i, Z^i) \;=\; &\frac{1}{2} \big[ \mathrm{Tr}(Z^{i\top}(L_{T_A}^\top L_{T_A} - D_{T_A})Z^i) + \mathrm{Tr}(D_{T_A} Z^i U^i) \\
&+\; \mathrm{Tr}(Y^{i\top} L_{T_A}^\top L_{T_A}) - 2\,\mathrm{Tr}(Z^{i\top} L_{T_A}^\top L_{T_A} Y^i) \\
&+\; \mathrm{Tr}(Z^i (L_{T_B}^\top L_{T_B} - D_{T_B})Z^i) + \mathrm{Tr}(D_{T_B} Z^i U^i) \\
&+\; \mathrm{Tr}(Y^i L_{T_B}^\top L_{T_B}) - 2\,\mathrm{Tr}(Z L_{T_B}^\top L_{T_B} Y^i) \big].
\end{aligned}
\tag{12}
$$

This loss is sound for alignments problems since, when $Y_i$ and $Z^i$ are in $\mathcal{Y}$, this is simply the $\ell_S$ loss $\|L_{T_A} Y_i - L_{T_A} Z^i\|_F^2 + \|Y_i L_{T_B} - Z^i L_{T_B}\|_F^2$.

Thus we get the following overall dual objective:

$$
\begin{aligned}
\min_{(Z^1,\dots,Z^n) \in \overline{\mathcal{Y}}} \quad & \frac{1}{2\lambda n^2} \| \sum_{i=1}^{n} -\sum_{j,k} (Y_i - Z^i)_{j,k}(a_j - b_k)(a_j - b_k)^T \|_F^2 \\
&-\; \frac{1}{n} \Big( \sum_{i=1}^{n} [\mathrm{Tr}(Z^{i\top}(L_{T_A}^\top L_{T_A} - D_{T_A})Z^i) + \mathrm{Tr}(Z^{i\top} D_{T_A} U^i) \\
&+\; \mathrm{Tr}(Y^T L_{T_A}^\top L_{T_A}^i) - 2\,\mathrm{Tr}(Z^{i\top} L_{T_A}^\top L_{T_A} Y^i) + \mathrm{Tr}(Z^i (L_{T_B}^\top L_{T_B} - D_{T_B})Z^{i\top}) \\
&+\; \mathrm{Tr}(U_i D_{T_B} Z^i) + \mathrm{Tr}(Y L_{T_B}^\top L_{T_B}^i) - 2\,\mathrm{Tr}(Z L_{T_B}^\top L_{T_B} Y^{i\top}) ] \Big).
\end{aligned}
\tag{13}
$$

We recall that $D_T$ is a diagonal matrix such that $L_T^\top L_T - D_T \preceq 0$ and thus our objective is convex. Our dynamic programming algorithm (DTW) is able to maximize any linear function over the sets

$\overline{\mathcal{Y}}(X_i)$. Thus we can use a Frank-Wolfe [2] algorithm. At iteration $t$, this algorithm iteratively computes a linearization of the function at the current point $(Z^1, \dots Z^n)_k$, computes a linearization of the function, optimize it, get a new point $(Z^1, \dots Z^n)_k^\star$ and then make a convex combination using a stepsize $\gamma$.

Note that we have directly a stochastic version of such an algorithm. As noted in Lacoste-Julien et al. [3], instead of computing a gradient for each block of variable $Z^i$, we simply need to choose randomly one block at each timestep and make an update on these variables.

The linearization simply consists in computing the matrix gradient for each of the matrix variables $Z^i$ which turns out to be:

$$
\begin{aligned}
\nabla_{Z^i}(g) \quad = \quad & \frac{1}{n}\Big[C - \frac{1}{2}\big(2(L_{T_A}^\top L_{T_A} - D_{T_A})Z^i + D_{T_A}U_i - 2L_{T_A}^\top L_{T_A}Y^i \\
+ \quad & 2Z^i(L_{T_B}^\top L_{T_B} - D_{T_B}) + U_i D_{T_A} - 2Y^i L_{T_A}^\top L_{T_A}\big)\Big]
\end{aligned}
\tag{14}
$$

where $C$ is simply the affinity matrix of dynamic time warping.

## 2 The dynamic time warping algorithm

Let us give the pseudocode of the dynamic time warping that maximize the LP (2) of the article. In opposition to Müller [4], we give a version of the algorithm for the affinity matrix $C$. Intuitively, the cost matrix is the opposite of a cost matrix, thus we aim to maximize the cumulated affinity instead of minimizing the cumulated cost. This corresponds exactly to the case of the article where we aim at maximizing a linear form over the set $\mathcal{Y}(X)$. This algorithm has complexity $O(T_A T_B)$, making it very costly to compute for large time series.

*Computing the cumulated affinity matrix D:*
$T_A, T_B \leftarrow \mathrm{size}(C)$
$D \leftarrow \mathrm{zeros}(T_A + 1, T_B + 1)$
**for** $i = 1$ **to** $T_A$ **do**
   $D(i, 0) \leftarrow -\infty$
**end for**
**for** $j = 1$ **to** $T_B$ **do**
   $D(0, j) \leftarrow -\infty$
**end for**
**for** $i = 1$ **to** $T_A$ **do**
   **for** $j = 1$ **to** $T_B$ **do**
      $D(i, j) \leftarrow C(i, j) + \max(D(i - 1, j), D(i, j - 1), D(i - 1, j - 1))$
   **end for**
**end for**
*Backtracking:*
$Y \leftarrow \mathrm{zeros}(T_A, T_B)$
$i \leftarrow T_A$
$j \leftarrow T_B$
**while** $i > 1$ **or** $j > 1$ **do**
   $Y(i, j) \leftarrow 1$
   **if** $i = 1$ **then**
      $j \leftarrow j - 1$
   **else if** $j = 1$ **then**
      $i \leftarrow i - 1$
   **else**
      $m \leftarrow \max(D(i - 1, j), D(i, j - 1), D(i - 1, j - 1))$
      **if** $D(i - 1, j) = m$ **then**
         $i \leftarrow i - 1$
      **else if** $D(i, j - 1) = m$ **then**
         $j \leftarrow j - 1$
      **else**
         $i \leftarrow i - 1$
         $j \leftarrow j - 1$
      **end if**
   **end if**
**end while**
**return** $Y$

Figure 1: The dynamic time-warping algorithm that solves the LP (2), for a given similarity matrix $C$.