[Reviews · NeurIPS 2014]

Submitted by Assigned_Reviewer_2

This paper proposes an algorithm to learn a distance metric for time series alignment. The proposed method falls into the structured output prediction framework, and is solved by a combination of convex optimization and dynamic programming. The method is evaluated on synthetic and realistic audio alignment tasks, and demonstrates significant improvement over baseline methods. Overall, this paper presents an interesting method for a real problem faced by practitioners dealing with time-series alignment tasks.

The paper is generally well written and easy to follow, although a few points could be stated more clearly. The general approach is novel and makes intuitive sense. The experimental evaluation is reasonably thorough, but the specific selection of features used in the experiments seems out of step with standard approaches (see below).

Technical comments and questions for the authors:

- As stated in the supplementary material, the relaxation of the output space to its convex hull may result in non-integral solutions. However, the text does not clearly describe how this is dealt with in practice to produce feasible solution points. Please expand and clarify this point.

- The experimental evaluation uses standard spectral features (mfcc, spectral centroid, etc), but curiously avoids commonly used features for musical sequence alignment (chroma or constant-Q spectra). In order to accurately gauge the potential of this method for improving alignment accuracy, the experiments should be expanded to better represent current alternative methods.

- The notation on L and L1 could be clarified

- eq. 6 has an extra parenthesis

- line 321 is missing a period

- (supplement) eq. 6 is missing a non-negative projection

- (supplement) line 93: is A'A supposed to be L'L?
Summary: This paper proposes a structured output prediction approach to learning a distance metric for optimizing multi-dimensional sequence alignment. The proposed method is novel and interesting, and the paper is generally well-written.

Submitted by Assigned_Reviewer_13

This paper presents a technique to learn a similarity measure for the problem of aligning temporal sequences. The authors combine several existing ideas and develop a technique for an audio problem that has practical significance.The paper is clear, well motivated. A few comments:

- Section 1 - It would be useful to mention a speech application as these ideas are quite relevant to speech.

- Section 5.1 - Some of the features that you use here are individual MFCCs. This is not reasonable as an MFCC vector (typically 13 dimensions) is meant to be used as a whole (as you have later done). An individual MFCC is should not be expected to work well and is similar to using a single value in a SIFT vector for a vision problem. Therefore, it would be better to show a single pair of bars in Figure 3 when running the expt. using an MFCC vector rather than separately on individual MFCCs.

- Section 5.3 - It would be useful to show some metrics that quantify “similar”. I understand that there is not enough space for a table, but mentioning the numbers would be useful.

Minor typo
- alignement -> alignment
- consists in -> consists of
Summary: This is an interesting paper that combines existing ideas to solve an audio problem of practical significance.

Submitted by Assigned_Reviewer_33

The paper proposes an approach for learning a Mahalanobis metric for sequence alignment using a structured output prediction framework. It proposes to exploit a symmetrized version of the area loss between two alignments and shows how to make its optimization tractable.

The paper is roughly well written but could be improved. Few forward references (e.g. using notations that have not been already defined) make a linear reading difficult. Also few claims that I list below are not so clear would maybe require more comments.
- Page 4 : I am not sure why if one signal is a reference \alpha would be steadily increasing, which is required for the area loss to be linear in Y.
- Page 5 : Second formulation of the symmetrized loss, defined on \bar{Y}(X): The fact that you work on this second formulation of the loss means that you optimize on \bar{Y}(x) rather than on Y(X) ? On page 6 the optimization is indeed performed on \bar{Y}. How to you get a solution in Y at the end?

The experimental section show improved results when using your approach (using l_S) enabling the combination of features on the two datasets that you use. I have yet a few comments.

On the first dataset you do not provide the results of an approach using uniform loss (i.e. using squared Euclidean distance instead of Mahalanobis), and it seems that you do not provide the best results from [14] either. Maybe because your experiments are not fully comparable with that of [14]? Also it seems you get much different results on this dataset as the ones published in [14] when looking at the accuracies reached with isolated features. You should probably add a discussion on these differences. Also why don’t you provide any comparative results of l_O?

On the second dataset, you almost reach the performance of the best single handcrafted feature with the combination of your standard features. I understand it is already a good result but would your approach be able to reach even better results by combining all these handcrafted features?

More generally I am not fully convinced by the experiments which miss alternative baselines. Also if one compares such alignment scores to what is learned in Hidden Markov models (HMMs) the learning of the Mahalanobis metric appears as a very narrow topic. Indeed learning a Mahalanobis distance in a Dynamic Time Warping alignment procedure would correspond to using a shared covariance matrix for all Gaussian distribution in a HMM, which is known to be a poor choice. From this viewpoint the topic of the paper, although it is relevant and the proposed method is smart and principled, appears of moderate importance.

Minor remarks
- Bottom page 2 : Last term in bottom of the page one guesses what #Y_{t,s} is as well as what s is but it is not introduced.
- Caption of Figure 1.
o Red upper -> red lower
o Blue lower -> blue upper
o \delta_{abs} is undefined at this point.
- Caption of figure 2
o “the most violated constrained” -> “the most violated constraint”
o You say first that “we have represented the most violated constrained at the end of learning, when the training loss is the Hamming one or the symmetrized area loss” where these two cases are indeed much different on the figure. And then you say “Note also that, in terms of Hamming loss the most violated contraint for l_S and the Hamming one are the same, while they differ much for the area loss.” which seems contradictory with the first sentence.
- Equation 6 : missing ^2 for the first norm term
- Page 8 : missing part in the sentence : “Note also that, in terms of Hamming loss the most violated contraint for l_S and the Hamming one are the same, while they differ much for the area loss.”
Summary: The paper proposes an approach for learning a Mahalanobis metric for sequence alignment using a structured output prediction framework. It proposes to exploit a symmetrized version of the area loss between two alignments and shows how to make its optimization tractable.

The proposed method is relevant and is original. Yet although the experimental section show improved results of the proposed approach there is no extensive comparison with competitive methods so that the experiments are not fully convincing.
Author Feedback
Author rebuttal: We thank the reviewers for their constructive comments and their careful reading of our work. We will correct minor typos and forward references for the final version if accepted.

Learning formulation:

- With the symmetrized area loss, loss-augmented inference is not done over \mathcal{Y} anymore but over its convex hull \bar{\mathcal{Y}}. Thus there is no need for rounding, which would not be a simple task.

- We acknowledge that the expression “taking a signal as reference” is confusing. We mean that one of the signals, let us say X_1, is a template on which we aim at aligning the other, let us call it X_2. Namely, for every index i, there is exactly one index j such that X_2(j) is aligned with X_1(i). In particular, note that the length of X_1 is necessarily smaller than the length of X_2.
This is the case in the audio-to score alignment literature.

- As mentioned by Rev. 1, the ideas of our work are relevant to speech recognition. DTW is used in this field since its introduction in the work of Sakoe and Chiba Dynamic Programming Algorithm Optimization for Spoken Word Recognition.

- Concerning the comparison with HMMs models for the task we consider, we are not aware about competitive methods using HMMs for audio-to-audio alignment. But we acknowledge that generative models are widely used in the context of audio-to-score or audio to partition (as in [12]), and for speech alignment tasks.
Note also that our metric learning is discriminative and that we are not fitting a covariance matrix of a generative model.
To overcome the limitations of using a temporally uniform weighting over features, we plan to make the metric depending on the position within the signals to align.

Experiments:

- As suggested by Rev. 3, we conducted an experimental comparison using the uniform loss (standard Euclidean metric) using the handcrafted features. As expected, this leads to poor results. On the Chorales dataset, we performed the alignment task using the plain Euclidean metric on the handcrafted features (simply taking W as the identity in Eq. 3). The results are off the charts on Fig. 3 with \delta_{\abs} at 4.1 seconds (individual values ranging from 1.4 seconds to 7.4 seconds) and \delta_{\max} at 8.7 seconds on average (individual values ranging from 4.2 seconds to 15.7 seconds).

- Rev. 1 pointed out the fact that we didn’t give the quantitative results for the best combination of MFCCs for the first dataset. We conducted this experiment whose results were the following: in terms of \delta_{\abs}, the best learned combination of MFCC is at 0.078 seconds and in terms of \delta_{\max}, the performance is at 0.46 seconds. These results are slightly worse than the best single handcrafted feature, but better than the best MFCC coefficient used as a feature. We will add these results to the final version of this work if accepted.
We want to stress out that all the corresponding results for the Chorales dataset are on Fig. 4 (combination of MFCC, best single feature, combination of handcrafted features).

- We agree with Rev. 1 that we could have used MFCCs as a whole unweighted vector. However, in our work, we mainly followed the approach of [14] which considers MFCCs as coefficients and not vectors, cf. Fig. 5 in [14] for instance.

- The performance of the individual features is more or less of the same order of magnitude as in the paper of [14] regarding \delta_{\abs}. A possible explanation for the changes in \delta_{\max} could lie in a difference of implementation. As the source code used in [14] is not available, because it is now used in a commercial software, we are not able to check this.

- We did not use the chroma features because they appear to be somewhat redundant with the MFCC features. Concerning the constant-Q spectra, we intended to use them but could not do so by lack of time.

- No comparative results with the loss l_O are provided by lack of time. Nevertheless, according to the intuition we have developed, l_O is a poorer approximation to the true area than l_S.